

# BAG3 regulates bone marrow mesenchymal stem cell proliferation by targeting INTS7

Yubo Liu[1,*], Renjie Xu[1,*], Jinfu Xu[2], Tiantian Wu[2] and Xiangxin Zhang[1]

[1] Department of Orthopaedics, Suzhou Municipal Hospital, The Affiliated Suzhou Hospital of Nanjing Medical University, Gusu School, Nanjing Medical University, Suzhou, Jiangsu, China
[2] State Key Laboratory of Reproductive Medicine, Department of Histology and Embryology, Nanjing Medical University, Nanjing, Jiangsu, China
[*] These authors contributed equally to this work.

## ABSTRACT

**Background**. BAG3 is an essential regulator of cell survival and has been investigated in the context of heart disease and cancer. Our previous study used immunoprecipitation-liquid chromatography-tandem mass spectrometry to show that BAG3 might directly interact with INTS7 and regulate bone marrow mesenchymal stem cell (BMMSCs) proliferation. However, whether BAG3 bound INTS7 directly and how it regulated BMMSCs expansion was unclear.

**Methods**. *BAG3* expression was detected by quantitative real-time PCR in BMMSCs after siRNA-mediated *BAG3* knockdown. BMMSC proliferation was determined using the CCK-8 and colony formation assays. The transwell migration, flow cytometry and TUNEL assays were performed to measure BMMSC migration, cell cycle and apoptosis, respectively. Moreover, co-immunoprecipitation, protein half-life assay and western blotting analyses were used to determine the regulatory mechanism underlying the BAG3-mediated increase in BMMSC proliferation.

**Results**. The results showed that knocking down BAG3 in BMMSCs markedly decreased their proliferative activity, colony formation and migratory capacity, and induced cell apoptosis as well as cell cycle arrest. Meanwhile, overexpression of BAG3 had the opposite effect. Bioinformatics and BAG3-INTS7 co-immunoprecipitation analyses revealed that BAG3 directly interacted with INTS7. Moreover, the downregulation of *BAG3* inhibited the expression of INTS7 and promoted its ubiquitination. We also observed that *BAG3* knockdown increased the levels of reactive oxygen species and the extent of DNA damage in BMMSCs. Notably, the upregulation of *INTS7* or the addition of an antioxidant scavenger could rescue the BMMSC phenotype induced by *BAG3* downregulation.

**Conclusions**. BAG3 directly interacts with INTS7 and promotes BMMSC expansion by reducing oxidative stress.

# INTRODUCTION

Bone marrow mesenchymal stem cells (BMMSCs) are characterized by high cell proliferation capacity and the ability to differentiate into adipocytes, hematopoietic

Corresponding author
Xiangxin Zhang,
lovezxx2003@njmu.edu.cn

progenitors, chondrocytes, and osteoblasts, which play essential roles in the progression of osteoporosis (*Ge & Zhou, 2021*; *Jiang et al., 2021*) and osteoarthritis (*Wei et al., 2021*). Specifically, the senescence and impaired osteoblastic differentiation of BMMSCs accelerate bone loss and microstructure degeneration during the pathogenesis of osteoporosis (*Jiang et al., 2021*). Pre-clinical studies of osteoporosis treatment *via* BMMSC transplantation have demonstrated successful and safe results. On transplantation, these BMMSCs prevented bone deterioration by stimulating osteogenesis and increasing bone mineralization. However, the self-renewal ability and activity of BMMSCs decreases with age (*Gavazzo et al., 2021*; *Stolzing et al., 2008*), which might lead to problems when performing BMMSC transplantation in older individuals. Moreover, the complex molecular network involved in BMMSC senescence is still largely unknown. Hence, new molecular markers and mechanisms related to BMMSC proliferation need to be identified to delay the senescence of BMMSCs and improve their efficacy in cell-based therapy.

The Bcl-2-associated athanogene 3 (BAG3), is a 575 amino acid (aa) multifunctional protein, which belongs to a family of co-chaperones that interact with Hsp70 *via* the BAG domain. This 110–124 aa BAG domain is highly conserved among Hsp70-beinding proteins (*Bienert et al., 2017*). BAG3 is an essential regulator for regulating the proteome and maintaining cellular survival (*Klimek et al., 2017*), and its role of BAG3 has been widely investigated in heart disease and cancer (*Kirk, Cheung & Feldman, 2021*). In the heart, BAG3 promotes autophagy, inhibits apoptosis, and maintains the structure of the sarcomere (*McClung et al., 2017*). In cancer cells, BAG3 binds to and supports multiple survival protein; therefore, it may be a promising therapeutic target (*De Marco et al., 2018*). Multiple previous studies have demonstrated that BAG3 interacts with Hsp70 to increase the expression of certain anti-apoptotic proteins and thus influence cancer cell survival and growth (*De Marco et al., 2018*). These findings indicate that BAG3, as an anti-apoptotic protein, participates in growth of multiple cell types, including normal and tumor cells. However, the effects of BAG3 on BMMSC growth, and the potential mechanisms implicated, remain unclear.

In our previous study, we found that INTS7, which interacts with the RNAP II subunit to participate in DNA damage response, adipocyte differentiation and maturation, hematopoiesis and tumorigenesis, is an essential regulator of BMMSC proliferation and osteogenic differentiation by directly interacting with ATP binding cassette subfamily D member 3 (ABCD3) (*Liu et al., 2021*). More importantly, *INTS7* downregulation increased reactive oxygen species (ROS) levels and DNA damage and decreased the expression levels of antioxidants in BMMSCs. This suggests that oxidative stress is involved in the regulation of INTS7/ABCD3-mediated BMMSCs proliferation (*Liu et al., 2021*). Using a combination of immunoprecipitation, liquid chromatography, and tandem mass spectrometry (IP-LC-MS/MS), we also found that additional co-factors, including BAG3, were linked to INTS7 function (*Liu et al., 2021*). However, it was unclear whether BAG3 and INTS7 worked together to regulate BMMSC proliferation and migration.

In this study, we aimed to investigate how BAG3/INTS7 co-regulatory axis regulates BMMSC proliferation and delineate the mechanism involved. We observed that *BAG3* knockdown or overexpression significantly impacted BMMSCs proliferation, apoptosis,

and colony formation ability. Notably, we also found BAG3 directly interacted with INTS7 and prevented its ubiquitination and degradation. Finally, we further explored whetheroxidative stress signaling was involved in the BAG3/INTS7-mediated induction of BMMSCs proliferation and migration.

## MATERIAL AND METHODS

### Cell culture and transfection

The OriCell Strain C57BL/6 mouse BMMSCs were purchased from Cyagen Biosciences (Guangzhou, China) and were cultured in OriCell C57BL/6 mouse BMMSC complete medium containing 10% fetal bovine serum (FBS), 1% glutamine, and 1% penicillin-streptomycin at 37 °C in a humid 5% $CO_2$ atmosphere. The phenotype, pluripotency, and stem cell properties of the BMMSCs were evaluated as previously reported (*Chen et al., 2020*; *Liu et al., 2013*; *Liu et al., 2021*). The BAG-targeting small interfering (si)RNA (si-BAG3) or the negative control siRNA (si-NC) were transfected into BMMSCs using Lipofectamine 2000 (Invitrogen, Waltham, MA, USA), according to the manufacturer's instructions. The nucleotide sequences of the BAG-targeting siRNAs were as follows: si-BAG3 #1, 5′-AAGAUGCAGUGUCCUUAGG-3′; si-BAG3 #2, 5′-UUGGCUUCUAGCUGUUGGC-3′. The nucleotide sequence of the si-NC was described previously (*Xue et al., 2022*). Additionally, the pcDNA3.1-BAG3, pcDNA3.1-INTS7, and empty vector (pcDNA3.1-NC) were purchased from GenePharma (Suzhou, China) and used to transfect BMMSCs in Opti-minimum essential medium (MEM; Invitrogen, Carlsbad, CA, USA) using the Lipofectamine 2000. Cells were harvested for analysis 48 h post-transfection.

### RNA extraction and quantitative real-time(qRT)-PCR

To further assess the expression of target genes in the transfected cells, total RNA was extracted from transfected cells using the RNeasy Plus Micro Kit (Qiagen, Düsseldorf, Germany), according to the manufacturer's instructions. A PrimeScript Reverse Transcription Kit (Vazyme, Nanjing, China) was then used for reverse transcription, and the SYBR Premix Ex Taq polymerase (Takara, Dalian, China) were used to conduct the qRT-PCR on an ABI 7500 Real-Time PCR System (Applied Biosystems, Foster City, CA, USA). *18S* ribosomal (r)RNA was used as an endogenous control and expression was calculated using the $2^{-\Delta\Delta Ct}$ method. The following primers were used in the qRT-PCR reaction: *BAG3* forward 5′-CCGGGCTGGGAGATCAAAAT-3′ and reverse 5′-GGCTGAAGATGCAGTGTCCTTA-3′; and *18S* rRNA forward 5′-AAACGGCTACCACATCCAAG-3′ and reverse 5′-CCTCCAATGGATCCTCGT TA-3′.

### Cell proliferation assays

Cell proliferation was detected using the Cell Counting Kit-8 (CCK-8) (Beyotime, Nantong, China) at four time points (0, 24, 48, 72, and 96 h), in accordance with the manufacturer's protocol. Briefly, after transfection for 48 h, 4,000 cells/well were seeded into 96-well plates. The CCK-8 solution was then added to each well and incubated at 37 °C for another 2 h. The optical absorbance of each well was read at 450 nm using a microplate reader (Molecular Devices LLC, Sunnyvale, CA, USA).

For the colony-formation assay, cells were plated in 6-well plates at a density of 1,000 cells/well and cultured for 2 weeks, before harvesting. Colonies were fixed with methanol and stained with 0.1% crystal violet dye (Beyotime). The number of cell colonies was observed and counted using a bright-field microscope.

## Transwell migration assay

Migration was assessed using the transwell migration assay, as reported previously (*Wang et al., 2022*; *Xu et al., 2022*; *Xue et al., 2022*; *Zhou et al., 2022b*). Briefly, after transfection for 48 h, $3 \times 10^4$ cells in 300 µL of serum-free medium were seeded into the upper chamber of each well of a 24-well plate (Merck Millipore), while 700 µL complete medium was added to the lower chamber. After culture for 24 h, cells in the lower chamber were collected, then fixed with 4% paraformaldehyde and stained with 0.1% crystal violet for counting.

## Cell cycle and apoptosis assays

For cell cycle analysis, BMMSCs were fixed in 70% ethanol at 4 °C for 48 h and then stained with propidium iodide (PI). Thereafter, the relative content of intracellular DNA was measured by flow cytometry (BD Biosciences, Heidelberg, Germany).

Cellular apoptosis was assayed using an Apoptosis Detection Kit (Vazyme), as previously mentioned (*Xue et al., 2022*; *Zhao et al., 2019*; *Zhou et al., 2022a*). Briefly, cells were fixed in methanol and treated with 0.1% Triton X-100 for 10 min at room temperature, followed by BrightRed Labeling Buffer for 60 min at 37 °C. After washing with phosphate buffered saline (PBS) three times, the samples were stained with DAPI. Images were captured using a confocal laser microscope (Zeiss LSM800; Carl Zeiss, Oberkochen, Germany).

## Co-immunoprecipitation (Co-IP) assay

Co-IP assay was performed by incubating cell lysates with an anti-BAG3 antibody (Proteintech, Chicago, IL, USA) or an anti-INTS7 antibody (Proteintech), as previously described (*Xue et al., 2022*). Briefly, the cells were lysed on ice for 30 min using RIPA lysis buffer (Beyotime, Nantong, China) and then precleared with protein A agarose(Invitrogen). The lysates were incubated with the indicated antibodies overnight at 4 °C and then with protein A agarose at 4 °C for 3 h. The immunoprecipitants were washed with PBS, eluted with sodium dodecyl sulfate buffer, and analyzed by western blotting.

## Protein-protein interaction

BAG3 and INTS7 sequences (FASTA format) were obtained from the National Center for Biotechnology Information (NCBI). Homology modeling analysis was done through the Robetta Modeling Server (https://robetta.bakerlab.org). The output is a three-dimensional (3D) protein model in Protein Data Bank (PDB) format. The Robetta server identified hot residues with 79% accuracy by calculating parameters such as implicit solventization and hydrogen bonding, stacking interactions, solventization interactions, and Lennard-Jones interactions. We then predicted the BAG3-INTS7 interaction interface using HawkDock and visualized it using PyMOL (https://pymol.org/2/).

## Western blotting

Western blotting was performed as previously described (*Xue et al., 2022*; *Yu et al., 2022*). In brief, cell lysates were isolated on an SDS-PAGE gel and then transferred onto polyvinylidene difluoride membranes(Millipore, Bedford, MA, USA). The membranes were blocked in Tris-buffered saline tween (TBST) containing 5% fat-free milk for 1 h at room temperature, and incubated with the indicated dilutions of primary antibodies (anti-BAG3, 1:500; anti-INTS7, 1:1,000; and anti-tubulin, 1:1,000; all from Proteintech) at 4 °C overnight. The next day, the membranes were washed and treated with horseradish peroxidase (HRP)-conjugated secondary antibodies at room temperature for 1 h. Western blots were developed and visualized using the Enhanced Chemiluminescent Substrate (Thermo Scientific, Waltham, MA, USA). Tubulin was used as a loading control. For protein half-life assay, BMMSCs was incubated with 100 μg/mL cycloheximide (CHX) to block protein synthesis, and proteins were assayed after 0, 4 and 8 h.

## ROS analysis

After transfection, cells were incubated with 20 μM 2′, 7′-dichlorofluorescein diacetate (DCFDA; Invitrogen, Waltham, MA, USA) for 40 min at 37 °C. DCFDA is a fluorogenic dye, which is used for the measurement of hydroxyl, peroxyl, and other ROS activity within cells. Fluorescence intensity was observed using a confocal laser microscope (Zeiss LSM800; Carl Zeiss), according to our previously reported methods (*Liu et al., 2021*).

## Statistical analysis

GraphPad Prism 9.0 (GraphPad, San Diego, CA, USA) and SPSS 22.0 software (IBM, Chicago, IL, USA) were used for data analysis. All experiments were repeated at least three times. All quantitative data are expressed as the mean $\pm$ standard deviation (SD). The Student's $t$-test and the one-way ANOVA were used to calculate $p$-values. All $p$-values were two-sided, and the threshold for statistical significance was set at $p < 0.05$.

# RESULTS

## BAG3 promotes BMMSCs expansion *in vitro*

To investigate the effects of BAG3 on BMMSCs growth, we transfected BMMSCs with two BAG3-targeting siRNAs (si-BAG3 1# and si-BAG3 2#) to inhibit BAG3 expression. The qRT-PCR and western blots results showed that the expression of BAG3 was significantly lower in BMMSCs transfected with the two BAG3-targeting siRNAs, than in BMMSCs transfected with the negative control siRNA (si-NC) (Figs. 1A–1C). As expected, we observed a significant decrease in the proliferation and colony formation capacity of the BMMSCs transfected with si-BAG3, but not those transfected with si-NC, in the CCK-8 and colony formation assays (Figs. 1D–1F). Moreover, the transwell migration assay results showed a significant decrease in the number of migrating BMMSCs after *BAG3* downregulation (Figs. 1G and 1H). To further assess whether BAG3 silencing in BMMSCs had an effect on the cell cycle and the rate of their apoptosis, a flow cytometry and TUNEL analysis was performed. Flow cytometry exhibited that BAG3 silencing delayed the progression of cell cycle and inhibited cell proliferation by arresting BMMSCs at G0/G1

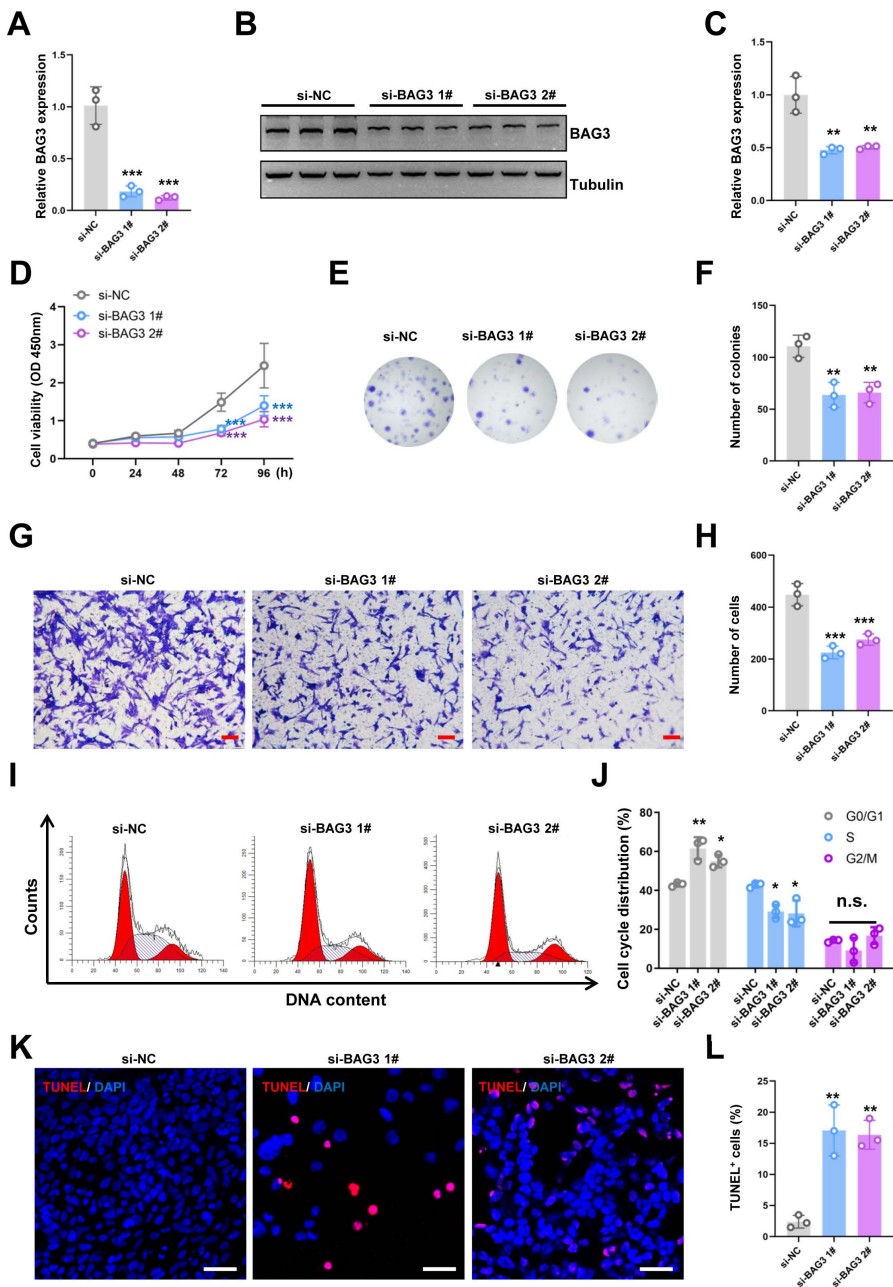

**Figure 1** **Knockdown of *BAG3* inhibits the expansion and increases the apoptosis of BMMSCs.** BMM-SCs were transfected with BAG3-siRNA (si-BAG3) to knockdown *BAG3*. BMMSCs transfected with the negative control siRNA (si-NC) were used as negative controls. (A) The relative mRNA expression of *BAG3* in BMMSCs transfected with si-BAG3 #1/#2 or si-NC was detected by qRT-PCR. (B–C) Western blot assays showed that the expression of BAG3 was decreased in BMMSCs transfected with BAG3 specific siRNA. (D) The CCK-8 assay was used to determine the viability of BMMSC following *BAG3* silencing. (E, F) The colony-forming ability of BMMSCs after transfection with si-BAG3 or si-NC. (G, H) The transwell migration assay was performed to investigate the migratory abilities of BMMSCs following *BAG3* (continued on next page...)

**Figure 1 (...continued)**
silencing; scale bar = 100 μm. (I, J) Flow cytometry for cell cycle in BMMSCs. (K, L) The TUNEL assay
was performed to detect the apoptosis of BMMSCs after *BAG3* knockdown; scale bar = 50 μm. Each ex-
periment was repeated three times, and the results are presented as the mean ± SD. $*p < 0.05$, $**p < 0.01$,
and $***p < 0.001$. Scale bar = 20 μm.

phase (Figs. 1I and 1J). We also found that the reduction in BAG3 expression increased the
number of apoptotic BMMSCs (Figs. 1K and 1L).

To further validate the role of BAG3 in BMMSCs, cells were transfected with pcDNA3.1-
BAG3 or vector control. Western blot assays showed that the expression of BAG3 was
increased in BAG3 overexpression cells (Figs. 2A and 1B). Moreover, cell growth was
markedly increased after *BAG3* overexpression in BMMSCs (Figs. 2C–2E). Meanwhile,
an increase in cell migration was observed following the transfection of BMMSCs with
pcDNA3.1-BAG3, compared with the migration of BMMSCs transfected with empty vector
(Figs. 2F and 1G). These data collectively indicate that BAG3 promoted the expansion and
migration of BMMSCs, while inhibiting their apoptosis.

## BAG3 interacts with INTS7

In our previous study, the endogenous INTS7 complexes purified from the BMMSCs
by IP were analyzed by MS (*Liu et al., 2021*) The results showed that BAG3 was a
potential interaction partner of INTS7. In the present study, we first predicted the
three-dimensional structure of the BAG3-INTS7 complex using the Robetta web
server (https://robetta.bakerlab.org/) (*Yang et al., 2020*) (Fig. 3A). We then predicted the
BAG3-INTS7 interaction interface using HawkDock and visualized it using PyMOL
(https://pymol.org/2/) (*Weng et al., 2019*) (Fig. 3B). The key residues predicted to be
involved in the BAG3-INTS7 interaction are listed in Table S1. Next, a BAG3-INTS7
Co-IP experiment was performed to support these bioinformatics models. The results
confirmed the protein interaction between BAG3 and INTS7 in BMMSCs (Fig. 3C). CHX
treatment reveals a significant reduction in INTS7 protein half-life after BAG3 interference
(Figs. 3D and 3E). Additionally, the western blotting results showed that *BAG3* silencing
reduced INTS7 protein expression, while treatment of MG132(proteaosomal inhibitor)
could rescue destabilized INTS7 expression following BAG3 knockdown (Figs. 3F and 3G).
These results indicate that BAG3 was indeed directly binding to INTS7 and affecting its
stability.

Ubiquitination marks proteins for degradation (*Van Wijk et al., 2019*). To investigate
whether ubiquitination was involved in the BAG3-mediated regulation of INTS7 stability,
we conducted ubiquitination experiments. As shown in Fig. 3H, a significant increase in
INTS7 ubiquitination was detected following the transfection of BMMSCs with si-BAG3
but not si-NC. These results show that BAG3 stabilized INTS7 and prevented it from being
marked for degradation by ubiquitination.

## *BAG3* silencing triggers oxidative stress

ROS regulate the survival, proliferation, and terminal differentiation of mesenchymal stem
cells (MSCs) (*Liu et al., 2021*). In general, low levels of ROS promote MSC proliferation,

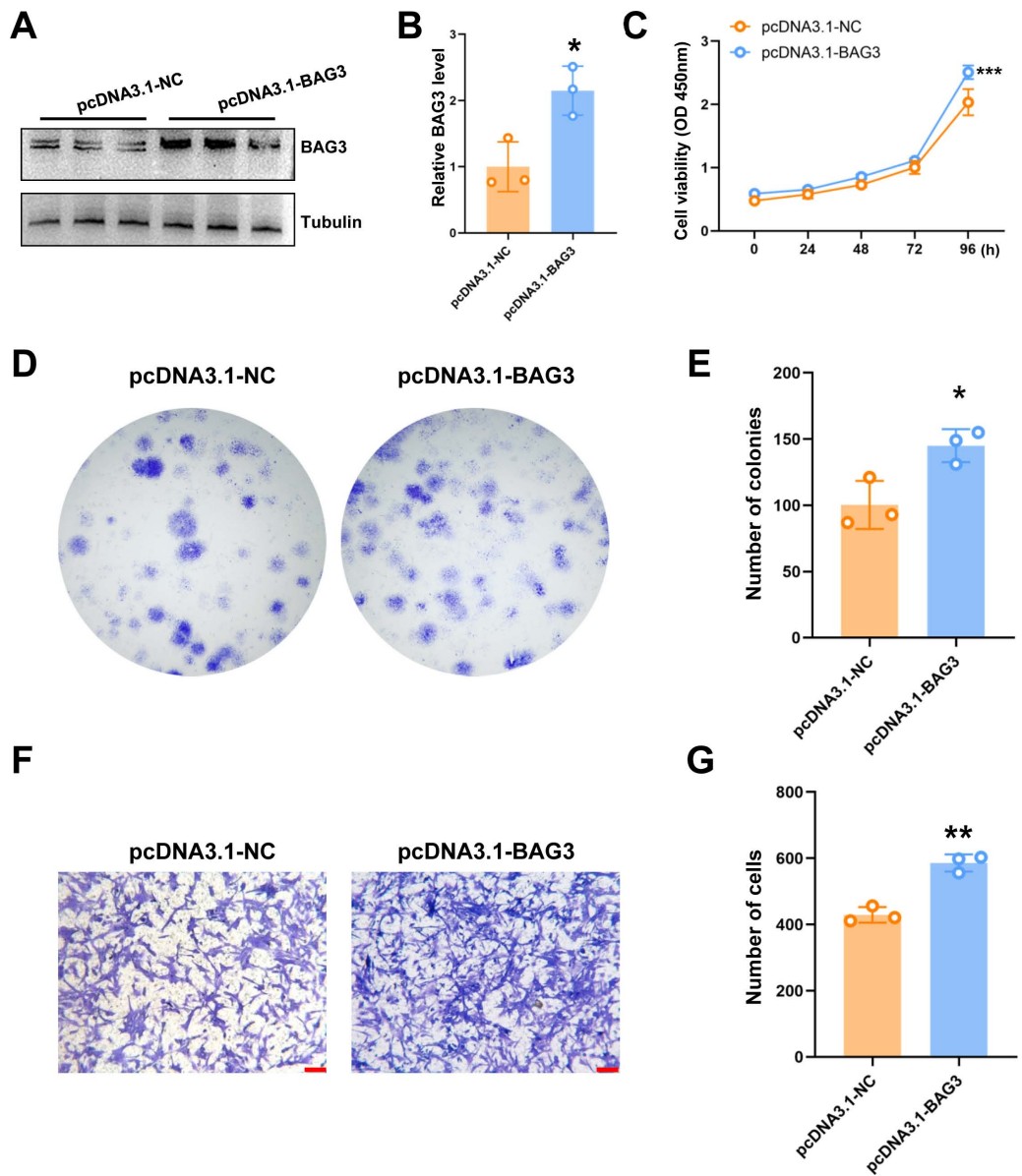

**Figure 2** **The overexpression of BAG3 promotes the proliferation and migration of BMMSCs.** BMM-SCs were transfected with pcDNA3. 1-BAG3 to overexpress BAG3. BMMSCs transfected with pcDNA3.1-NC were used as negative controls. (A-B) Western blot assays showed that the expression of BAG3 was increased in BAG3 overexpression cells. (C) CCK-8 and (D, E) colony formation assays were performed to determine the proliferation ability of pcDNA3.1-BAG3-transfected BMMSCs. (F, G) The transwell migration assay was performed to determine the migration ability of pcDNA3.1-BAG3-transfected BMMSCs or controls; scale bar = 100 μm. Values are expressed as means ± SD. *$p < 0.05$, **$p < 0.01$, and ***$p < 0.001$.

differentiation, and survival, while elevated ROS levels (defined as oxidative stress) induce MSC apoptosis. Moreover, our previous study reported that INTS7 and its binding protein ABCD3 affect ROS levels (*Liu et al., 2021*). We anticipated that BAG3, as an important binding partner of INTS7, could affect BMMSC proliferation *via* a similar mechanism. To

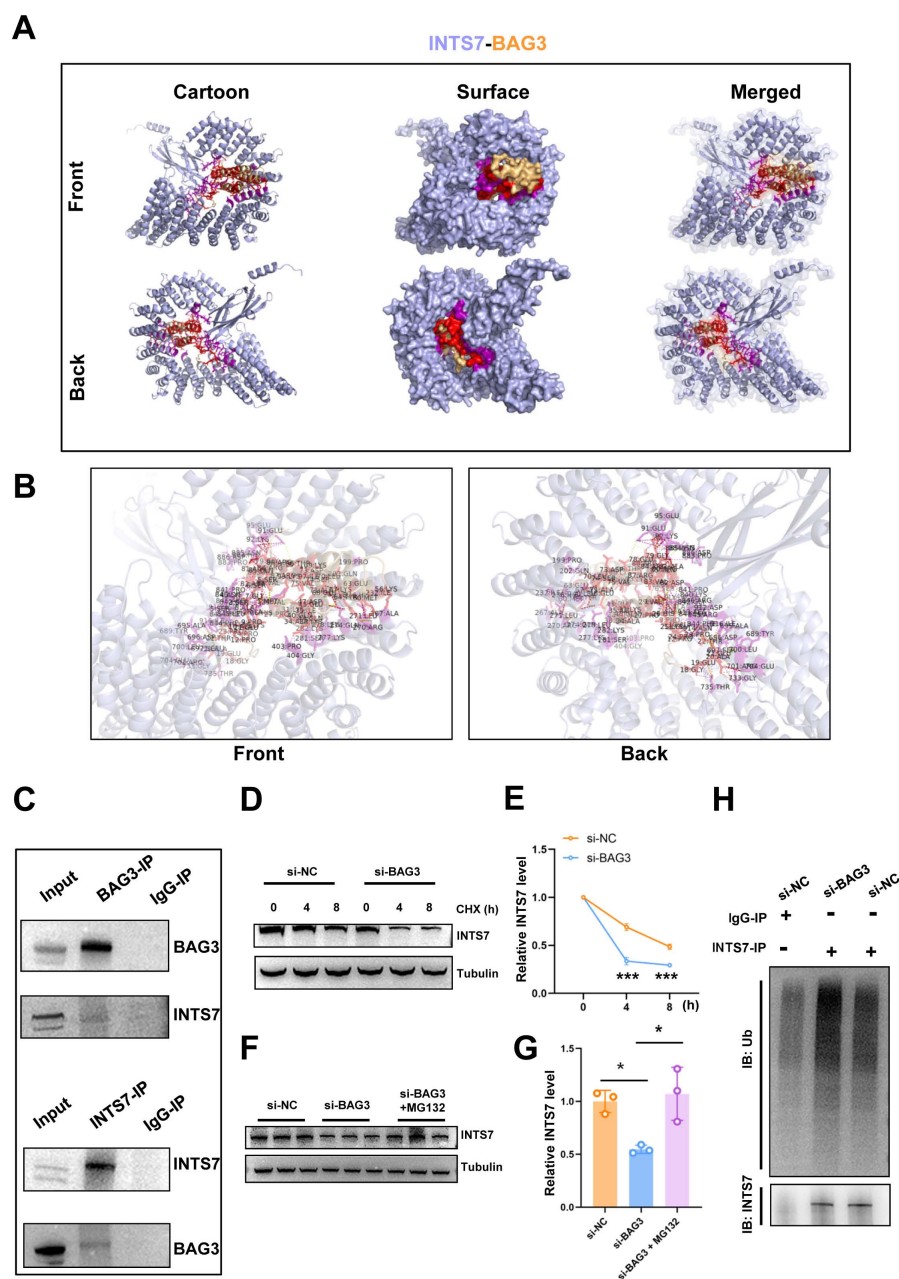

**Figure 3** **BAG3 interacts with INTS7.** (A) Modelling of the INTS7-BAG3 interaction. The "cartoon" mode represents the backbone as well as the secondary structures of the corresponding proteins. The "surface" mode displays the solvent-accessible surface area; BAG3 is shown in orange, INTS7 in blue, BAG3-binding sites in red, and INTS7-binding sites in purple. (B) The interface view shows the binding residues (in stick mode) at the interface between BAG3 and INTS7. The interacting residues are colored in red and purple for BAG3 and INTS7, respectively. The molecular contacts between interacting residues are connected with dashed yellow lines. (C) The co-immunoprecipitation assay using BMMSC lysates with anti-BAG3 and anti-INTS7 antibodies. (D–E) The expression of INTS7 were determined by western blotting after CHX treatment at the indicated times. (F) Western blotting analysis of INTS7 expression after the

**Figure 3 (…continued)**
transfection of BMMSCs with si-NC or si-BAG3. MG132 was used at the concentration of 20 $\mu$M. (G) Quantification of (F). (H) BMMSCs were transfected with si-BAG3 or si-NC and lysed with immuno-precipitation lysis buffer containing protease inhibitor. INTS7 was immunoprecipitated with anti-INTS7 antibody and detected with anti-ubiquitin (Ub) and anti-INTS7 antibodies. Analyses were performed in triplicate. $*p < 0.05$, $**p < 0.01$, and $***p < 0.001$.

investigate whether oxidative stress inhibition was involved in the BAG3-INTS7-mediated regulation of BMMSCs expansion, we examined ROS and antioxidant levels in BMMSCs after *BAG3* silencing. In addition, DNA damage is associated with oxidative stress induction. The detection of nuclear $\gamma$-H2AX foci, a biomarker for DNA double-strand breaks, provides indirect evidence of oxidative stress (*Sangermano et al., 2019*). We found that both the ROS levels (Figs. 4A and 4B) and the percentage of $\gamma$-H2AX-positive cells (Figs. 4C and 4D) were all significantly increased in BMMSCs transfected with si-BAG3 *vs.* si-NC. These findings suggest that oxidative stress was activated in BAG3-deficient BMMSCs.

## ROS regulate the BAG3-INST7-induced proliferation and migration of BMMSCs

To further investigate whether ROS altered the phenotype of BMMSCs with reduced BAG3 expression, we conducted rescue experiments by co-transfecting BMMSCs with pcDNA3.1-INTS7 and the antioxidant scavenger, N-acetylcysteine (NAC), respectively. As expected, pcDNA3.1-INTS7 and NAC reversed the effect of *BAG3* silencing on the proliferation (Fig. 5A), and colony formation (Figs. 5B and 5C) and migration capacities of BMMSCs (Figs. 5D and 5E). These results imply that ROS levels regulate the INST7-BAG3-mediated expansion of BMMSCs.

## DISCUSSION

In this study, we found that BAG3 interacts with INTS7 and controls BMMSC expansion. Our previous study reported that INTS7 affects ROS and antioxidant levels, as well as the extent of DNA damage, which all impact on BMMSC proliferation (*Liu et al., 2021*). We therefore surmised that BAG3, as a binding partner of INTS7, could play an essential role in BMMSC proliferation and apoptosis by reducing oxidative stress. To this end, we knocked down *BAG3* expression in BMMSCs, which markedly decreased their proliferative, colony formation, and migration ability, while inducing cell apoptosis. Meanwhile, the overexpression of BAG3 had the opposite effect, which suggested that BAG3 might help stem cells to overcome senescence. To the best of our knowledge, these results for the first time suggest that BAG3 is an essential regulator of BMMSCs proliferation.

Previous studies have indicated that BAG3 overexpression resulted in the resistance of tumors to apoptosis. BAG3 stabilizes pro-survival Bcl-2 family proteins (*e.g.*, Bcl-xL, Bcl-2, and Mcl-1), which are essential regulators of apoptosis (*Galluzzi et al., 2018*). In addition, BAG3 activates several key drivers of tumorigenesis, including the NF-$\kappa$B pathway, the cell-cycle regulator p21, and the FoxM1 transcription factor. It was suggested that BAG3 is also involved in tumor cell survival, stemness, and proliferation (*Kögel et al., 2020*). These results highlight the heterogenous mechanisms used by BAG3 to regulate cell survival.

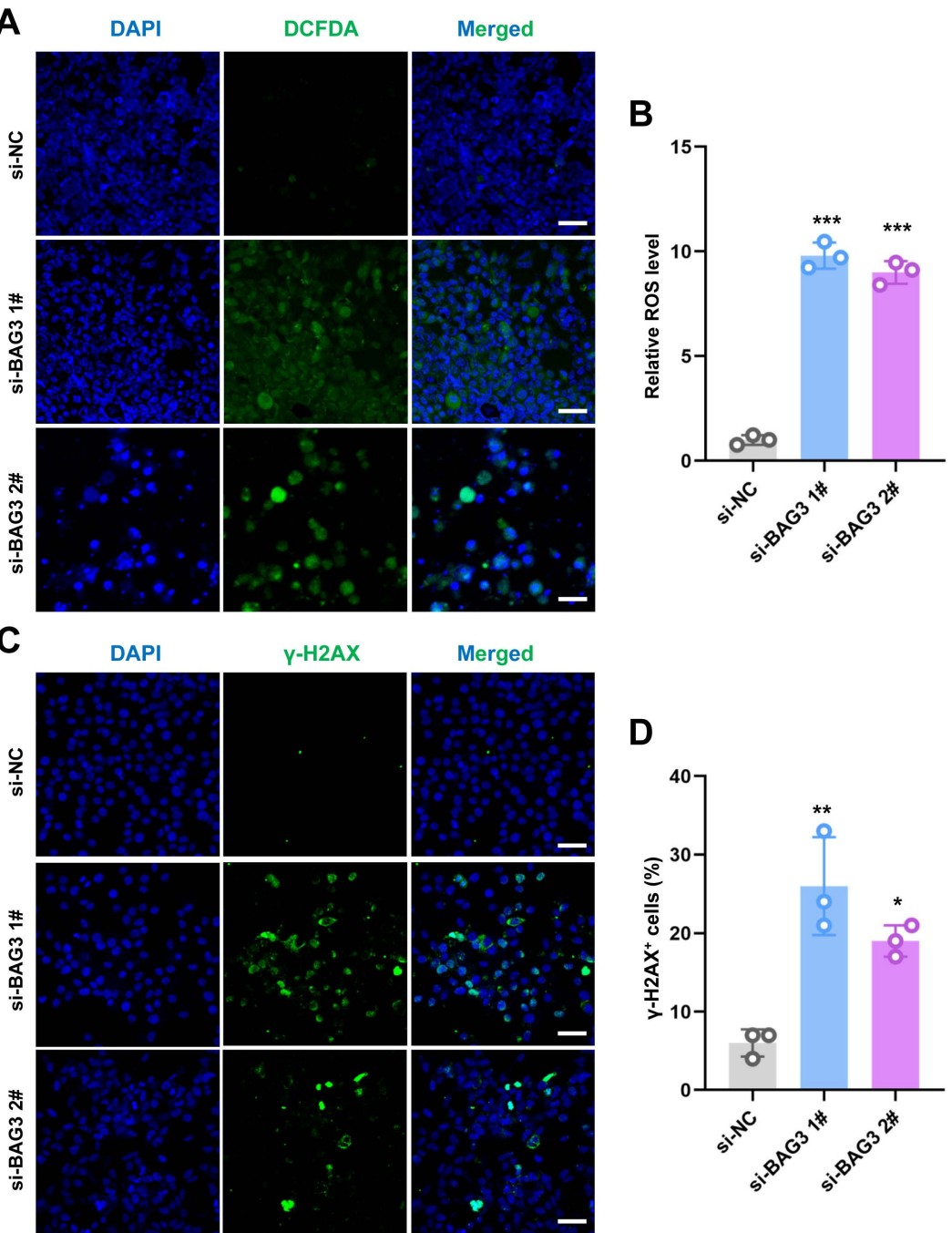

**Figure 4** **Knockdown of *BAG3* increases oxidative stress and DNA damage.** (A) Immunostaining of 2′,7′-dichlorofluorescein diacetate (DCFDA) in BMMSCs transfected with si-BAG3 or si-NC; scale bar = 50 μm. (B) Quantification of (A). (C) Immunostaining of $\gamma$-H2AX in BMMSCs transfected with si-BAG3 or si-NC; scale bar = 50 μm. (D) Quantification of (C). *$p < 0.05$, **$p < 0.01$, and ***$p < 0.001$.

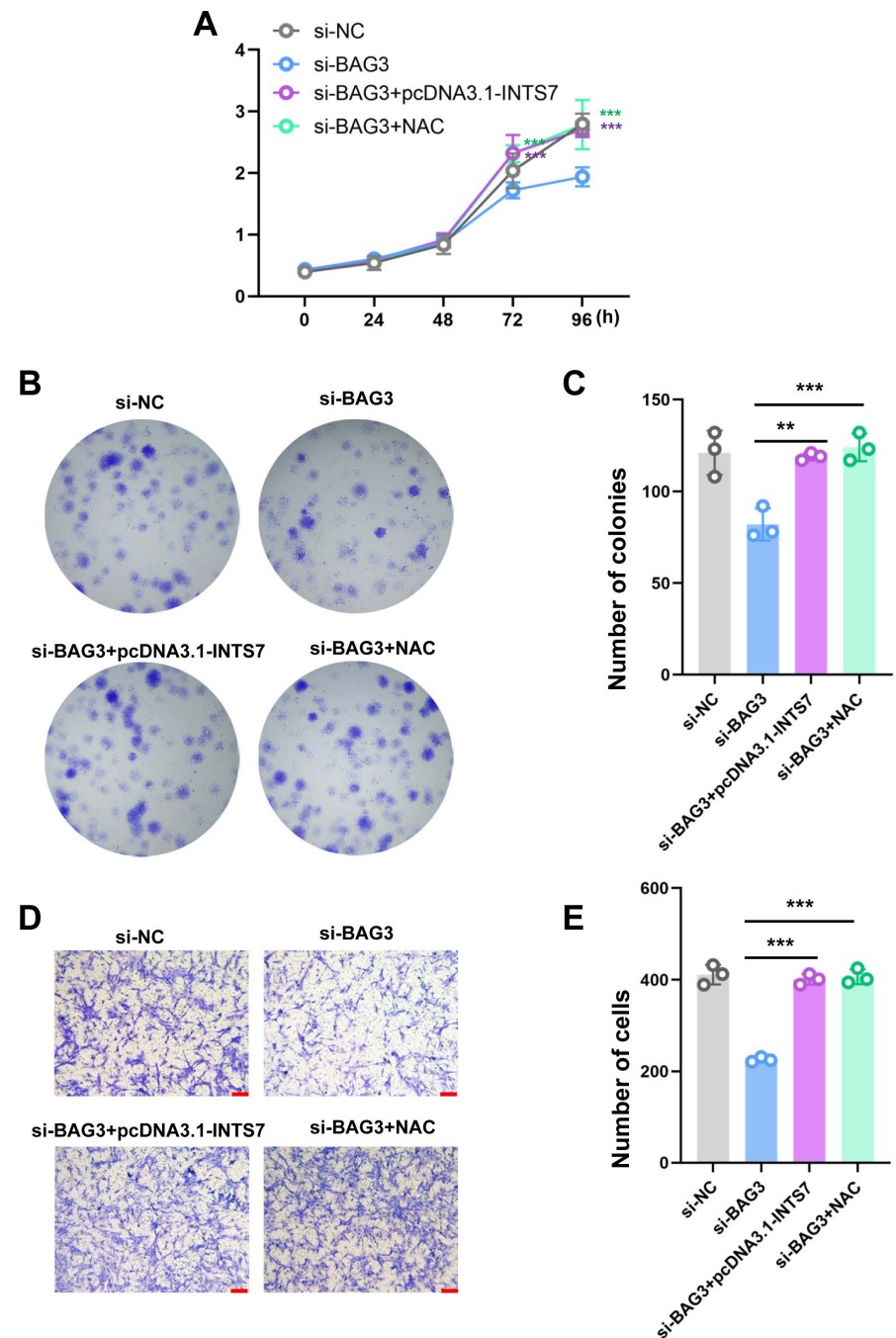

**Figure 5 Reactive oxygen species are involved in the INST7-BAG3-mediated regulation of BMMSC expansion.** (A) CCK-8 and (B, C) colony formation assays were performed to determine the cell viability and colony formation capacity of BMMSCs co-transfected with si-BAG3 and pcDNA3.1-INTS7 and treated with the antioxidant scavenger, N-acetylcysteine (NAC). (D, E) The transwell migration assay was performed to investigate changes in the migratory abilities of BMMSCs co-transfected with si-BAG3 and pcDNA3.1-INTS7 and treated with NAC; scale bar = 100 μm. $*p < 0.05$, $**p < 0.01$, $***p < 0.001$.

Our previous study used IP-LC-MS/MS to suggest that BAG3 interacts with INTS7 and could therefore affect BMMSC proliferation and survival (*Liu et al., 2021*). In the present study, we used bioinformatics analysis and Co-IP to confirm that BAG3 interacts with INTS7. We also observed that the downregulation of *BAG3* inhibited the expression of INTS7 and promoted its ubiquitination. It has been previously shown that *INTS7* downregulation increases ROS levels and decreases antioxidant levels in BMMSCs. In the present study, we observed that the downregulation of *BAG3* also increased ROS levels and increased the extent of DNA damage, suggesting that oxidative stress is activated in BAG3-deficent BMMSCs. Moreover, the silencing of BAG3 expression reduced the proliferation, and colony formation and migration abilities of BMMSCs. By contrast, increasing INTS7 expression or adding NAC rescued the BMMSC phenotype induced by *BAG3* downregulation. Collectively, our results provide new insights into the relationship between BAG3, INTS and BMMSC expansion. However, a limitation of this study was that we focused on a single mechanism by which BAG3 regulates BMMSC expansion; BAG3 may regulate BMMSC function in other ways, which will require further exploration.

In summary, we showed that BAG3 expression increased the proliferation, and the colony formation and migration abilities of BMMSCs *in vitro*. We also provided evidence that BAG3 interacts with INTS7 and protects it from degradation *via* ubiquitination. Moreover, we showed that oxidative stress was involved in the BAG3-INTS7-mediated regulation of BMMSCs. Thus, our data provided novel mechanistic insights into the function of BAG3 in BMMSCs proliferation.

### Funding
This work was supported by the Gusu Health Talent Program of Suzhou (Grant Number: GSWS2020058) and the Suzhou Science and Technology Development Plan (Grant Number: KJXW2021034). The funders had no role in study design, data collection and analysis, decision to publish, or preparation of the manuscript.

### Grant Disclosures
The following grant information was disclosed by the authors:
Gusu Health Talent Program of Suzhou: GSWS2020058.
Suzhou Science and Technology Development Plan: KJXW2021034.

### Competing Interests
The authors declare there are no competing interests.

### Author Contributions
- Yubo Liu performed the experiments, analyzed the data, prepared figures and/or tables, and approved the final draft.
- Renjie Xu performed the experiments, analyzed the data, prepared figures and/or tables, and approved the final draft.

- Jinfu Xu analyzed the data, authored or reviewed drafts of the article, and approved the final draft.
- Tiantian Wu analyzed the data, authored or reviewed drafts of the article, and approved the final draft.
- Xiangxin Zhang conceived and designed the experiments, authored or reviewed drafts of the article, and approved the final draft.

## Data Availability

The raw measurements are available in the Supplemental Files.

## Supplemental Information

Supplemental information for this article can be found online at http://dx.doi.org/10.7717/peerj.15828#supplemental-information.

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
