# Peer review of "BAG3 regulates bone marrow mesenchymal stem cell proliferation by targeting INTS7"

_PeerJ, doi:10.7717/peerj.15828_

## Round 0.1 · original submission · Major Revisions

Please address the concerns of the reviewers and amend the manuscript accordingly.

·

Basic reporting

Manuscript Summary: In this manuscript, Liu, Y et al, have evaluated the role of BAG3 in BMMSC proliferation, apoptosis and colony formation assay. In addition, the authors show that BAG3 interacts with INTS7 and impedes its ubiquitination. In addition, the authors also show that BAG3 knockdown induces oxidative stress resulting in DNA damage that can be rescued by INST7. The experimental design is straightforward, and the results are clear. However, some additional clarifications and details need to be added to make the results more compelling. Here are some of the concerns with regards to the manuscript.
1. The hypothesis in the Introduction section is not clear and as a result, the significance of the results presented is unclear.
2. In line 110, the authors have directly introduced INTS7 as an essential regulator of BMMSC proliferation and osteogenic differentiation. Although, this article is a continuation of the already published data, it would be advisable to give a brief introduction on what is INTS7 (please include the complete nomenclature of this protein) and its essential function in stem cell biology?

Experimental design

The experimental design is straightforward however there are areas where additional clarifications are needed to make the results more compelling. Here are some of the concerns with the experimental design:
1. The authors showed modeling of INTS7-BAG3 interaction in Figure 3A. The authors need to show the methodology used for modeling in method section of the manuscript so that it is easier for the targeted audience to review.
2. In line 252, the authors say that INTS7 directly interacts with BAG3. Although, the modeling and IP suggest interaction between the two proteins, it is still unclear if it would be direct interaction as the experimental data presented is only Co-IP. So it should rephrased in line 252.
3. In addition, the authors also suggests that BAG3 affect INTS7 stability in line 252 and 253. It is not clear from Fig 3F, the impact of BAG3 on INTS7 stability. I recommend that that the authors conduct a cycloheximide chase to show BAG3 knockdown does impact INST7 stability and secondly, does treatment of MG132 (proteaosomal inhibitor) rescue destabilized INTS7 expression following BAG3 knockdown. These experiments should be considered to be included in the manuscript.
4. In line 323, the authors suggests that BAG3 is a potential therapeutic target for BMMSC senescence. It is unclear from the data presented any relationship between BAG3 and cellular senescence as there are characterized protein markers used to classify senescence.
5. In addition, it is not clear if BAG3 knockdown have any functional role in regulating differentiation of BMMSC and/or if it’s major role is in maintaining proliferation of the stem cell pool.
6. Can overexpression of BAG3 help stem cells to overcome senescence? This can be addressed in the discussion section.

Validity of the findings

1. The significance of the findings need to be reported in detail.
2. In line 323, the authors have shown that BAG3 can be a potential target in musculoskeletal diseases associated with BMMSC senescence. However, the authors have not addressed how BAG3 expression change with age, and its relation to senescence (will a stable knockdown of BAG3 induce BMMSC to senescence and impede its differentiation potential?).
3. With respect to point 2 above, how can BAG3 be used as a therapeutic target. The authors need explain in detail and avoid generalized discussion points.

Additional comments

No additional comments.

Reviewer 2 ·

Basic reporting

This study builds upon previous research on INTS7 and investigates the direct binding of BAG3 to INTS7, as well as the mechanism behind how BAG3 regulates the expansion of bone marrow mesenchymal stem cells (BMMSCs). The findings reveal that BAG3 plays a crucial role in regulating BMMSC proliferation by targeting INTS7 directly. Both the overexpression and knockdown of BAG3 have been found to significantly impact BMMSC expansion, providing valuable insights into the underlying mechanisms of stem cell regulation.

Experimental design

The experiment has a good design.

Validity of the findings

The experiment has a good design, but it requires more detailed verification to strengthen its soundness. Both the overexpression and knockdown of BAG3 should be confirmed through Western Blotting and Immunofluorescence, provided that the necessary antibodies are accessible. These verifications could be included in Figure 1 and Figure 2. Flow cytometric analysis is also needed to determine the cell-cycle distribution of BM-MSCs. This would provide crucial information about the effects of BAG3 on cell proliferation and further enhance the validity of the experiment.

---

## Round 0.2 · Minor Revisions

Please address the remaining issue pointed out by the reviewer and make the proposed changes to Figure 3.

·

Basic reporting

The manuscript meets all the editorial requirements. There are only few minor edits required throughout the manuscript.

Experimental design

The experimental design is sound. I would advise the authors to consider another WB of better resolution for Figure 3H.

Validity of the findings

The findings support the conclusions drawn by the authors.

Additional comments

No Additional Recommendations.

Reviewer 2 ·

Basic reporting

No comment

Experimental design

NO comment

Validity of the findings

The questions were all addressed by author

---

## Round 0.3 · accepted · Accept

The remaining issues pointed out by the reviewer were adequately addressed and the revised manuscript was amended accordingly. Therefore, the revised version is acceptable now.